# Inappropriate Prescriptions in Older People—Translation and Adaptation to Portuguese of the STOPP/START Screening Tool

**DOI:** 10.3390/ijerph19116896

**Published:** 2022-06-04

**Authors:** Luís Monteiro, Matilde Monteiro-Soares, Cristiano Matos, Inês Ribeiro-Vaz, Andreia Teixeira, Carlos Martins

**Affiliations:** 1CINTESIS—Centre for Health Technology and Services Research, Faculty of Medicine, University of Porto, 4099-002 Porto, Portugal; matsoares@med.up.pt (M.M.-S.); inesvaz@med.up.pt (I.R.-V.); andreiasofiat@med.up.pt (A.T.); carlosmartins20@gmail.com (C.M.); 2USF Esgueira +, 3800-322 Aveiro, Portugal; 3MEDCIDS—Department of Community Medicine, Information and Decision in Health, Faculty of Medicine, University of Porto, 4099-002 Porto, Portugal; 4Escola Superior de Saúde da Cruz Vermelha Portuguesa, 1300-125 Lisbon, Portugal; 5Department of Mathematics, University of Aveiro, 3810-193 Aveiro, Portugal; cristiano.r.matos@gmail.com; 6Escola Superior de Tecnologia da Saúde de Coimbra, Instituto Politécnico de Coimbra, 3045-093 Coimbra, Portugal; 7Porto Pharmacovigilance Centre, Faculty of Medicine, University of Porto, 4099-002 Porto, Portugal; 8IPVC—Instituto Politécnico de Viana do Castelo, 4900-347 Viana do Castelo, Portugal

**Keywords:** geriatric medicine, quality in health care, general medicine, STOPP/START

## Abstract

Inappropriate prescribing, which encompasses the prescription of potentially inappropriate medications (PIMs) and potential prescribing omissions (PPOs), is a common problem for older people. The STOPP/START tool enables general practitioners, who are the main prescribers, to identify and reduce the incidence of PIMs and PPOs and appraise an older patient’s prescribed drugs during the diagnosis process to improve the clinical care quality. This study aimed to translate and validate the STOPP/START screening tool to enable its use by Portuguese physicians. A translation-back translation method including the validation of the obtained Portuguese version was used. Intra- and inter-rater reliability and agreement analyses were used in the validation process. A dataset containing the information of 334 patients was analyzed by one GP twice within a 2-week interval, while a dataset containing the information of 205 patients was independently analyzed by three GPs. Intra-rater reliability assessment led to a Kappa coefficient (*κ*) of 0.70 (0.65–0.74) for the STOPP criteria and 0.60 (0.52–0.68) for the START criteria, considered to be substantial and moderate values, respectively. The results of the inter-rater reliability rating were almost perfect for all combinations of raters (*κ* > 0.93). The version of the STOPP/START criteria translated into Portuguese represents an improvement in managing the medications prescribed to the elderly. It provides clinicians with a screening tool for detecting potentially inappropriate prescribing in patients older than 65 years old that is reliable and easy to use.

## 1. Introduction

Today, it is globally accepted that adverse drug reactions (ADRs) are a public health problem and have a significant clinical impact related to morbidity and mortality, which results in the increased use of health services in developed countries [1,2]. ADRs are responsible for about 7% of all hospital admissions, many of which are considered preventable [2,3]. Additionally, about 2–3% of patients admitted with an ADR die as a result [2,4].

ADRs may occur in 6–20% of patients admitted to hospitals, increasing their hospitalization period; highly increasing the costs associated with healthcare [5]; indirectly impacting patients’ and their families’ economic, social context, and psychological wellbeing [6,7]; and leading to the discussion of patient participation and involvement in pharmacovigilance [8,9].

The number of older adults is increasing [10], as is their life expectancy [11,12], and these patients are more likely to have more than one chronic disease [13,14] and be prescribed multiple drugs, increasing their susceptibility to inappropriate medication use [15,16,17,18,19,20,21].

Inappropriate prescribing that encompasses potentially inappropriate medications (PIMs) and potential prescribing omissions (PPOs) is a common problem for older people and is closely related to adverse events and ADRs [22]. Older adults are more prone to drug-related problems, as most take several medicines for multiple comorbidities, described as polypharmacy [23,24].

It is necessary to reduce PIMs and PPOs and improve clinical care quality [25]. The STOPP/START (Screening Tool of Older Persons’ Prescriptions—STOPP; Screening Tool to Alert to Right Treatment—START) criteria for the use of potentially inappropriate medication in older people recognize the dual nature of inappropriate prescribing by including a list of PIMs (STOPP criteria) and PPOs (START criteria).

STOPP/START is a valid, reliable, and comprehensive screening tool that enables the prescribing physician to appraise an older patient’s prescribed drugs in the context of their diagnosis [26]. Since the first publication of the STOPP/START criteria in 2008 [26], the tool has been widely disseminated and validated in many countries at different levels of healthcare (primary care, hospitals, nursing homes). The latest version (version 2) was published in 2014 and consists of 114 criteria, including 80 STOPP criteria and 34 START criteria [20,27]. These criteria are based on an up-to-date literature review and consensus validation among a European panel of experts [20]. The STOPP/START criteria were translated and adapted from English into several languages such as Czech, French [28,29,30], Italian, Spanish [31,32], and Dutch [33] to facilitate the local application of the criteria worldwide and have had a positive impact on patient evaluation [26].

This tool identifies potentially inappropriate prescriptions (PIPs) [34,35], identifying potentially inappropriate medicines through the STOPP criteria and identifying potential prescription omissions through the START criteria. The prevalence of patients with at least one instance of PIP identified by the STOPP criteria ranges from 21% [36] to 79% [37]. However, this range should be interpreted cautiously due to the heterogeneity of the sample population and study design between the different studies where this tool was assessed. The START criteria have identified at least one instance of PPO in 23% [36] to 74% [37] of patients.

A recent comparison of tools used to identify PIMs showed that the STOPP version 2 criteria identified substantially more PIMs than the EU (7)-PIM list [38], PRISCUS—Potentially Inappropriate Medications in the Elderly list [39,40,41], FORTA [39,40], and Beers criteria [25,42,43,44,45,46]. The STOPP/START criteria were found to be significantly associated with detecting adverse events in acutely ill older people, unlike the Beers criteria [20,42,43,44,45]. Compared to the Beers criteria or the prescribing indicators provided in the Elderly Australia criteria, the number and scope of drug-related problems identified were found to be best represented by the STOPP/START criteria [20,47,48]. Another advantage of this tool is that it considers PIMs and the indications to start an appropriate medication (START) [18].

A previous study from Gallagher et al., concluded that the STOPP/START criteria are generalizable across different European countries and languages [49]. Despite this, in other countries, such as in those with resource-limited healthcare settings, the original STOPP/START criteria might not be directly applicable; thus, modified versions of the STOPP/START criteria have been developed and validated recently [24]. In Portugal, this tool has already been used by Portuguese authors, but the translation and adaptation of the criteria have never been carried out, and the original tool is still used [25,50,51,52,53]. The current study aimed to translate and validate the STOPP/START screening tool to enable its use by Portuguese general practitioners/family physicians.

## 2. Materials and Methods

This study was conducted in four phases: The first phase (phase I) was the translation and adaptation of the STOPP/START screening tool to the Portuguese language, followed by patient data collection (phase II). Phase III consisted of an intra-rater reliability and agreement study, and phase IV consisted of an inter-rater reliability and agreement study. Pre-registration on the ‘Open Registries Network’ was conducted (DOI 10.17605/OSF.IO/SK2RJ (accessed on 31 March 2021), and the translation and adaptation of the STOPP/START screening tool to Portuguese has been described elsewhere [18].

### 2.1. Phase I: Translation and Adaptation of the STOPP/START Screening Tool to the Portuguese Language

The translation and adaptation of the STOPP/START screening tool followed the Principles of Good Practice for the Translation and Cultural Adaptation Process for Patient-Reported Outcomes Measures [30]. The adaptation and translation were carried out based on the 2014 O’Mahony et al., version of STOPP/START [20]. Permission from the STOPP/START’s authors to translate, adapt, and validate this tool for use in Portuguese was obtained by email. The final version was distributed to 15 general practitioners to verify if there were any interpretation issues and improve clarity. The research team analyzed the results obtained from applying the STOPP/START tool and prepared the final version. As a translation was needed, the chance for possible disagreements between raters was reduced by validating these translations before studying the intra-rater and inter-rater agreements. The detailed procedure was published previously in the protocol [18].

### 2.2. Phase II: Data Collection of Patient Data

Patients were randomly selected from a list of patients aged > 65 years old from a primary care center in the Centre Region of Portugal, following which a total of 8165 patients were followed, with 1625 aged over 65 years old. The sample size was calculated in the published protocol, and 334 subjects were randomly selected to participate in the study [26,54]. Exclusion criteria included incapacity or unwillingness to provide written informed consent, diagnosis of psychotic disorder, institutionalization, and the presence of terminal illness. Patients were interviewed during previously scheduled medical appointments. Every participant signed a written consent form (Appendix A). The identity of all participants was protected throughout the study.

Sociodemographic data such as age, sex, and educational level were collected and are shown in Table 1. Clinical data were collected by consulting health record registries and conducting interviews of clinical patients, including the identification of the total number of medications used for chronic diseases, any prescribed drugs, dosage, pharmaceutical dosage, pharmaceutical form and route of administration, reason for taking medication, allergies, drug-related conditions, history of adverse drug reactions, and current or past conditions/diseases. Other clinical information was also collected and described in the protocol but not used in the adaptation of the STOPP/START tool [18]. The information collected was input into a database, each patient was numbered from 1 to 334 by the main investigator, and the record of the coding was stored offline in an Excel 2016^®^ spreadsheet. All data recorded during this study will be stored for 5 years after the closure of the investigation, following the Portuguese Clinical Research Law. After this period, data containing participant codes will be destroyed.

### 2.3. Phase III: Intra-Rater Reliability and Agreement Study

As previously proposed by Kottner et al. [55], reliability may be defined as the ability of a measurement to differentiate among subjects or objects, comprising the ratio of variability between subjects or objects to the total variability of all measurements in the sample [56,57]. Intra-rater agreement assesses the extent to which the two responses from the same rater are concordant [58]. By definition, intra-rater reliability refers to the consistency of data recorded by the same rater, using the same scale, classification, instrument, or procedure, to assess the same subjects or objects at different times one rater over several trials. It is best determined when multiple trials are administered over a short period [55]. An independent researcher physician (named investigator/rater ‘A’) applied the Portuguese version of the STOPP/START criteria to all patient data collected in phase II. The investigator/rater ‘A’ was a family doctor with more than 10 years of experience in primary care, which included caring for and making daily prescriptions for older adults. To ensure intra-rater reliability and agreement, two weeks later, investigator A re-administered the tool. Both assessments of rater ‘A’ were used to study the intra-rater reliability.

### 2.4. Phase IV: Inter-Rater Reliability and Agreement Study

Inter-rater reliability refers to the consistency of data recorded by different raters, using the same scale, classification, instrument, or procedure, to assess the same subjects or objects. Inter-rater agreement assess the extent to which the responses of two or more independent raters are concordant [58]. In this specific study, intra-rater and inter-rater reliability assist in determining if the measurement tool produces results that can be used by a clinician to make decisions confidently [55]. Three independent researchers (named investigators/raters ‘B’, ‘C’, and ‘D’) independently applied the Portuguese version of STOPP/START using the data collected in phase II. The investigators/raters ‘B’, ‘C’, and ‘D’ were family doctors with more than 10 years of experience in primary care. For the total of 334 subjects who participated in the intra-rater study, 205 patients were randomly selected for the inter-rater assessment [59]. These three physicians were independent investigators and only had contact with the authors to access the collected data. These investigators independently assessed the STOPP/START criteria in each of the 205 patients and were invited to provide written comments if necessary. Inter-rater agreement was assessed by comparing the results of the three raters. Between raters ‘B’, ‘C’, and ‘D’, an inter-rater reliability test was performed. Inter-rater reliability assessment is useful because observers will not necessarily interpret answers (or tools) in the same way and may disagree on how the constructed tool is used [60,61].

### 2.5. Statistical Analysis

Data were stored with Microsoft Excel 2016^®^ software (Microsoft Corporation, Redmond, WA, USA). Data analyses were conducted using SPSS^®^ V.27.0 (SPSS Inc, Chicago, IL, USA) and R Studio^®^ V 1.3.1093 (Integrated Development for R. RStudio, PBC, Boston, MA, USA). Categorical variables were described using absolute and relative frequencies, *n* (%). Quantitative variables were summarized by means and their respective standard deviations (SDs), along with minimum and maximum values (min–max). According to the ‘Guidelines for reporting reliability and agreement studies’, reliability analyses and agreement analyses (intra- and inter-rater) were performed using Kappa statistics and the proportions of a specific agreement, respectively [62,63,64]. The Kappa statistics were interpreted as poor if the score was ≤0.2, fair if it was 0.21–0.40, moderate if it was 0.51–0.6, substantial if it was 0.61–0.8, and good if it was 0.81–1.00. The proportion of specific agreement distinguishes agreement on positive (PPos) or negative (Pneg) proportions, which might have different implications in clinical practice [65]. The 95% confidence intervals (95% CI) were presented for Kappa statistics and agreement proportions [55].

Kappa statistics were used for the calculation of both inter- and intra-rater reliability [66]. The Kappa statistic is a coefficient of reliability for categorical data [67]. As the Kappa coefficient is known to be affected by rare observations, it may not always reflect the true agreement rates and will provide an underestimation of the actual agreement [68]. A simple solution for this problem is calculating the proportions of agreement and separating the agreement rates into positive and negative agreements, thus making it easier for readers to interpret the results [63,69].

## 3. Results

A total of 334 patients were enrolled in this study. The patients’ characteristics (age, sex, educational level, and number of medicines used) are described in Table 1.

Educational level was grouped according to the International Standard Classification of Education (ISCED 2011) [70]. The number of medicines used was grouped according to the definition of polypharmacy, grouping the number of medicines [23,71,72,73].

Intra-rater reliability and agreement involved the analysis of Rater A’s evaluation of 334 patients’ records and re-evaluation after a 2-week interval. Results are reported in Table 2 (STOPP) and 3 (START). Inter-rater reliability and agreement analyses were performed by three different raters (‘B’, ‘C’, and ‘D’) who evaluated 205 randomized patients from the database. Each rater evaluated the same patients to allow for their comparison. The results obtained for the inter-rater reliability and agreement using the STOPP and START tools are shown in Table 2 and Table 3, respectively.

For the STOPP criteria, the intra-rater reliability showed a Kappa coefficient of 0.70 [95% CI 0.65–0.74], considered substantial; the positive and negative proportions of agreement obtained were 94.2% [95% CI 93.1–95.1] and 75.2% [95% CI 70.9–79.1], respectively. The results obtained for the inter-rater reliability were almost perfect, with *κ* near to one in all possible combinations of raters. Inter-rater agreement determines the agreement between pairs of raters and all raters’ judgments regarding the STOPP criteria.

For the START criteria, the intra-rater reliability showed a Kappa coefficient of 0.60 [0.52–0.68], considered a moderate value; the positive and negative proportions of agreement obtained were, respectively, 88.2% [85.4–90.6] and 71.1% [64.5–76.8]. The inter-rater reliability results were almost perfect, with κ near to one in all possible combinations of raters. Inter-rater agreement determines the agreement between pairs of raters and all raters’ judgments regarding the START criteria.

The final version of the Portuguese adaptation of STOPP/START is presented in Appendix A.

## 4. Discussion

This is to the best of our knowledge, the first study to translate and adapt the STOPP/START screening tool to Portuguese. The intra-rater reliability and inter-rater reliability scores obtained were not inferior to those obtained in previous studies conducted in other languages [28,29,30,31,32,33].

When testing reliability, several approaches are taken to determine consistency [74,75]. However, according to Innes et al., test–retest reliability, intra-rater reliability, and inter-rater reliability are the most common measures used among work-related assessments [74,75].

The first source of intra-rater inconsistency could be explained by various factors related to the assessment process. Rater A presented a high disagreement between two STOPP and START criteria evaluations. A major explanation was based on the analysis of the discrepancies. From 129 discrepancies seen between the first and second evaluation, 94 were related to proton pump inhibitors (F2 or A1 criteria without further investigation). In the second evaluation, with a better knowledge of the tool, the drug was properly assessed. Out of 119 discrepancies found in the evaluation on the START criteria, 51 were related to the introduction of vaccines in the second evaluation (I1 or I2 criteria were used). According to previous studies, a high level of familiarity is required to efficiently apply the STOPP/START criteria in clinical practice [49]. Additionally, raters could differ concerning their experience, specialties, and professional skills and have different perceptions regarding the knowledge required to use a particular item of the assessment tool. It is therefore important to highlight that the professionals that perform medication reviews with the STOPP/START tool should receive adequate training in order to use the tool appropriately [76,77].

One strength of this study is its innovation, with it representing the first development of a Portuguese version of the STOPP/START criteria. Our research was not merely a translation, but also an adaptation carried out by independent general practitioners that will hopefully increase the use of this version in the primary care setting. To ensure intra-rater reliability and agreement, the same doctor re-evaluated patients’ records by applying the same criteria 2 weeks later, avoiding recall bias. Additionally, this study provides evidence for a near-perfect inter-rater reliability, meaning that raters almost always agree on whether to exclude/include medicines, although the reasons for these decisions were not necessarily similar. Finally, this version translated into Portuguese can be used by general practitioners or any other medical practitioner and could be used in countries where Portuguese is the main language. However, the differences in healthcare systems between countries; the different ranges of medicines available; and differences in population characteristics, such as genetic or racial differences, should be considered.

One potential limitation was related to the fact that the adapted version of STOPP/START exclusively focuses on primary healthcare centers. The authors deliberately did not include patients with specific pathologies. It is important to clarify that this tool may not be appropriate for use in all population groups or in different healthcare settings, and the assessment tool should be evaluated in future studies, including in other populations with specific pathologies and in different contexts.

Furthermore, some of the randomized patients (*n* = 26, 8%) did not have any drugs prescribed, which would have reduced discrepancies between the raters when evaluating the STOPP criteria. Another potential limitation is the fact that the current tool was originally published in 2014, which means that there may be new medication and/or additional therapeutic indications that do not fit the current tool. Finally, the raters’ decision to stop or start a drug based on this tool was a dichotomous decision and was not validated as either right or wrong from a clinical point of view. No assessment of clarity was performed, so a quality appraisal study should be conducted in the future to improve the clarity of clinical practice guidelines on a language level and enhance its clinical applicability [78].

In addition, using this tool, raters can point out different reasons for withdrawing or adding drugs without this changing the final decision. Since the criteria were applied to data from files in the absence of a clinical evaluation of patients by raters and prescriptions are subject to a certain variation in interpretation concerning the clinical heterogeneity observed in the elderly population, clinical evaluation was not performed by PPOs and the reasons for stopping and starting drugs were not compared [79].

## 5. Conclusions

The major research result of the current study was the adaptation of the STOPP/START (2014) criteria into Portuguese.

The objective of our research was not to test the tool in a Portuguese population. The use of this tool in this context may not lead to clinical differences for patients, or, at least, this was not the main objective for its use in this study.

The STOPP/START criteria have been proven to be a good tool for detecting potentially inappropriate prescriptions and improving prescription quality in older people in all healthcare settings, therefore leading to improved quality of life in patients, reducing the incidence of PIMs and PPOs, and improving clinical care quality. This research provides clinicians with a screening tool with which to detect potentially inappropriate prescribing in patients older than 65 years old that is easier to use for Portuguese native speakers. The tool is also useful for improving the training of medical students in managing polypharmacy [76] and can have a positive economic impact by reducing medicine expenditure in older patients [80,81]. This version in Portuguese represents a step forward in improving the management of medications in the elderly. The adaption of this tool will be useful not only for Portugal but also for other Portuguese-language countries.

## Figures and Tables

**Table 1 ijerph-19-06896-t001:** Characteristics of patients (*n* = 334).

Variable	*n* (%)
**Age, years mean (SD); min–max**	74.2 (6.9); 65–99
65–69 years	105 (31.4%)
70–74 years	71 (21.3%)
75–79 years	78 (23.4%)
80–84 years	50 (15.0%)
85+ years	30 (9.0%)
**Sex**	
Women	159 (47.6%)
Men	175 (52.4%)
**Education level**	
Early childhood, primary and lower secondary education (level 0–2)	316 (94.6%)
Upper secondary and post-secondary non-tertiary education (levels 3–4)	17 (5.1%)
Short-cycle tertiary education, Bachelor’s, Master’s, or Doctorate (levels 5–8)	1 (0.3%)
**Number of medicines used**	
0–1	57 (17.1%)
2 to 4 (Minor polypharmacy)	66 (19.8%)
5 to 9 (Major polypharmacy)	210 (62.8%)
10+ (Severe polypharmacy)	1 (0.3%)

Discrepancies in totals are due to rounding.

**Table 2 ijerph-19-06896-t002:** Intra- and inter-rater reliability and agreement based on the analysis of the STOPP criteria.

STOPP Criteria	Rater Combination	Agreement (%)		Reliability
		Ppos † (95% CI)	Pneg + (95% CI)	Kappa (95% CI)
Intra-rater	Rater A Rater A	94.2 (93.1–95.1)	75.2 (70.9–79.1)	0.70 (0.65–0.74)
Inter-rater	Rater B Rater C	99.8 (99.4–99.9)	98.9 (97.1–99.7)	0.99 (0.97–1.00)
	Rater B Rater D	99.6 (99.1–99.8)	97.8 (95.5–99.1)	0.97 (0.95–0.99)
	Rater C Rater D	99.5 (99.1–99.8)	97.5 (95.0–98.8)	0.97 (0.95–0.99)
	Rater B Rater C Rater D	99.6 (99.3–99.8)	98.1 (96.6–99.2)	0.98 (0.94–1.00)

† Ppos, agreement on positive proportions. + Pneg, agreement on positive negative proportions.

**Table 3 ijerph-19-06896-t003:** Intra- and Inter-rater reliability and agreement from the analysis of the START criteria.

START Criteria	Rater Combination	Agreement (%)		Reliability
		PPos † (95% CI)	PNeg + (95% CI)	Kappa (95% CI)
Intra-rater	Rater A Rater A	88.2 (85.4–90.6)	71.1 (64.5–76.8)	0.60 (0.52–0.68)
Inter-rater	Rater B Rater C	98.7 (97.3–99.4)	94.4 (88.6–97.7)	0.93 (0.87–0.99)
	Rater B Rater D	98.7 (97.3–99.4)	94.4 (88.6–97.7)	0.93 (0.87–0.99)
	Rater C Rater D	99.1 (97.9–99.7)	96.1 (90.1–98.7)	0.95 (0.91–1.00)
	Rater B Rater C Rater D	98.8 (97.9–99.6)	94.9 (90.8–98.2)	0.94 (0.87–1.00)

† PPos, agreement on positive proportions. + Pneg, agreement on positive negative proportions.

## Data Availability

The datasets presented in this study are available on request from the corresponding author.

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
