# Peer review of "Inappropriate Prescriptions in Older People—Translation and Adaptation to Portuguese of the STOPP/START Screening Tool"

_ijerph, 2022, doi:10.3390/ijerph19116896_

Round 1

Reviewer 1 Report

This is a vey interesting article.

Some general comments follow:

STOPP_START(SS) v2  dates from 2014.  There may be new medications or medications with new indications since 2014.

Importance of adequate training to use SS should be emphasised.

Medication review one part of broader clinical assessment. Pharmacists with SS training valuable in clinical assessment of patients. (SS originally developed by doctors and pharmacists in Ireland).

SSv2 may not be appropriate for all population groups e.g. has not been validated in the population with intellectual disabilities.

Portuguese SSv2 may not be appropriate in all regions of the world where portuguese spoken  e.g. Africa, Oceania, Asia.  Different  healthcare systems, different range of medicines available, people with different genetic/racial characteristics.

Portuguese SSv2 suitable for primary care (line 244)... may not be appropriate in all healthcare settings.

No mention of improved quality of life in conclusions section.

Concern has been expressed in some articles concerning CLARITY of SSv2 in english.  Was there any feedback of the clarity of Portuguese SS?

Very worthwhile project.

Best of luck.

Some minor issues follow:

Line 42 word missing at end of line?

Table 1:   figures disaligned when printed 

Tables: 3 and 4 : these did not print out

Reviewer 2 Report

Here are some suggestions for the improvement of the current version of the manuscript:

  1. The entire article needs English editing.
  2. The Abstract needs to be re-written (especially the first 2 sentences).
  3. Table 1: data should be better aligned
  4. Please define what intra-/inter-rater reliability means
  5. Please revise lines 216 - 220

Reviewer 3 Report

This current study addressed an important issue of potentially inappropriate medication, and analyzed a valuable dataset of the Portuguese elderly in the primary health care setting. But the authors need to make the following revisions before the manuscript could be accepted for publication.
1. The explanations on STOPP/START need to be re-arranged in a logical way. In line 52-88 the authors used three paragraphs to explain the tool from different angles; however, the explanations were not arranged in a logically clear pattern and some were overlapped. The full name of STOPP/START was even missing when it first appeared in line 53. The possible presentation structure could be 1) what is STOPP/START, 2) why it is important, 3) how to use it and the current evidence of effect, 4) the lack of a Portuguese version.
2. This current study translated and validated STOPP/START for use in the Portuguese context. Since no change were made to the base 2014 version except for the language, it might be inappropriate to use ‘STOPP/START.v2’(line 270) and similar expression such as ‘version 2’ in the manuscript supplementary material.
3. In the section of 2.5. Statistical analysis, the authors do not need to explain the what are intra-rater and inter-rater reliability analyses such as line 159-163 and line 170-176. These explanations should be presented in section 2.3. and 2.4.
4. As not only reliability analyses were performed but also the agreement tests, the authors have to provide the statistical analyses methods for the agreement tests as well.
5. Line 216-218 and line 219-220 are contradictory, for there were no ‘Tables 4 and 5’ in the manuscript, and these lines explained the same issues but referred to different tables/supplementary material.
6. As a non-native Portuguese speaker and cannot read Portuguese in supplementary appendix I, I am reluctant to point out that the title in supplementary appendix I should be double-checked. But the number in the title attracted my attention, as the number ‘3’ looked strange in a material with the name of ‘appendix I’. I used an online translation tool and the results confirmed my doubt. Please check.
7. In line 217-218, the authors suddenly introduced the base 2014 version. It had better be removed to the Methods section.
8. In the Conclusions section, the authors need to first conclude the major research findings of this current study before discussing their future use. In line 272, the authors used the expression of ‘more logical, more reliable’, but did not provide the reference object.
9. The authors should make a thorough check of typos in the manuscript, e.g. 1) ‘ratter’ in line 94, 134 and 137, and 2) the format of references in the manuscript were not uniformed like ‘(52)’ in line 147.

Reviewer 4 Report

title can be improved 

need thorough review for language edit

on what basis 2-4 medicine is categorized as minor polypharmacy. please clarify or support with evidence

it is not clear that the content is to test the tool or to develop and validate the tool

Round 2

Reviewer 2 Report

The authors have adressed all my comments.

Reviewer 3 Report

The authors have made appropriate revisions according to last round's comments. It is a nice reading experience. Thank you very much.